

# Tagging moisture sources with Lagrangian and inertial tracers: Application to intense atmospheric river events

Vicente Pérez-Muñuzuri[1], Jorge Eiras-Barca[1], and Daniel Garaboa-Paz[1]

[1]Group of Nonlinear Physics. Faculty of Physics. University of Santiago de Compostela. 15782 Santiago de Compostela, Spain.

*Correspondence to:* V. Pérez-Muñuzuri (vicente.perez@cesga.es)

**Abstract.**

Two Lagrangian tracers tools are evaluated for studies on atmospheric moisture sources and pathways. In these methods, a moisture volume is assigned to each particle which is then advected by the wind flow. Usual Lagrangian methods consider this volume to remain constant and the particle follows flow path lines exactly. In a different approach, the initial moisture volume can be considered to depend on time as it is advected by the flow, due to thermodynamic processes. In this case, the tracer volume drag must be taken into account. Equations have been implemented and moisture convection was taken into account for both Lagrangian and inertial models. We apply these methods to evaluate the intense atmospheric rivers that devastated *(i)* the Pacific North West region of the United States, and *(ii)* the Western of the Iberian Peninsula, with flooding rains and intense winds in early November 2006, and May 20, 1994, respectively. We note that the usual Lagrangian method underestimates moisture availability in the continent while active tracers achieve more realistic results.

## 1 Introduction

Extreme precipitation and flooding in many areas of the world, and particularly on the US and Europe west coasts, are often related to the presence of atmospheric rivers (ARs) (*Dettinger et al.*, 2011; *Ralph et al.*, 2011; *Lavers and Villarini*, 2013; *Eiras-Barca et al.*, 2016; *Waliser and Guan*, 2017). ARs are narrow and elongated filamentous structures that transport moisture from the tropics into mid-latitudes over a period of a few days, and usually form in association with baroclinic systems (e.g. *Eiras-Barca et al.*, 2018). AR conditions occur in the warm sector of extratropical cyclones and are characterized by large water vapor contents and transport at low levels (*Ralph et al.*, 2004). For some AR events, a filamentous pattern develops persisting enough time to be considered a Lagrangian coherent structure (*Garaboa-Paz et al.*, 2015, 2017). ARs have been shown to play a key role in extratropical tropospheric dynamics (*Newel et al.*, 1992; *Zhu and Newell*, 1998; *Gimeno et al.*, 2016). The advection and convergence of moisture along ARs is a key process for Earth's sensible and latent heat redistribution and has a strong impact on the mid-latitudes water cycle by increasing tropospheric water vapor mixing (*Zhu and Newell*, 1998).

The present study examines two well-observed extreme precipitation events to better understand the role of the landfalling ARs that devastated portions of the US Pacific Northwest and Western of the Iberian Peninsula coasts with torrential rains and severe flooding on 6-7 November 2006 (*Neiman et al.*, 2008), and 20 May 1994 (*Lavers and Villarini*, 2013), respectively. The



composite analysis of the vertically integrated water vapor transport (IVT) and the integrated column of water vapor (IWV) (Figs. 1-2) provides a depiction of the landfalling AR during its most destructive phase. Both figures show a narrow plume extending northeastward from the tropical moisture reservoir to the Pacific Northwest of the United States and Western Iberian Peninsula, and strongly suggests direct incorporation of tropical moisture into the AR. For the 2006 event in the Pacific basin,

the high amounts of moisture transported by the AR can be partially explained by the tropical origin of the cyclone, which in conjunction with an anticyclone located to the southeast, increases the poleward and eastward flux of moisture along its track to mid-latitudes. For the 2004 event in the Atlantic basin, the transport of moisture is enhanced by the combined action of a cyclone located northwestern of the head of the AR and an anticyclone located southeastward of the AR. Landfalling AR events were observed to occur on 6th November 2006 and 20th May 1994, respectively, and led to important precipitation

amounts (Fig. 3) in the northwestern coast of North America and Iberian Peninsula.

To analyze the contribution of tropical moisture onto the landfalling ARs different numerical methods have been applied in the last decades, namely analytical, Lagrangian and Eulerian models (e.g. *Gimeno et al.* (2012), for a detailed review). Lagrangian models have been widely used in climatic studies of atmospheric water vapor sources and in the diagnosis of the origin of moisture in extreme precipitation events (*Stohl and James*, 2004, 2005; *Gimeno et al.*, 2010; *Ramos et al.*, 2016). These

15 models, although widely used, cannot describe correctly evaporation ($e$) and precipitation ($p$), in addition to neglecting liquid water and ice, which results in an overestimation of both $e$ and $p$. All Lagrangian models consider constant parcel volumes. However, the initial parcel may change its volume along its pathway due to thermodynamic and mechanical effects. In this case, inertial effects on the parcel should be considered (*Maxey and Riley*, 1983). Finite-size or inertial particle dynamics in fluid flows can differ markedly from Lagrangian particle dynamics, in both, their motion and clustering behavior (e.g. *Michaelides*

(2003), for a review). Eulerian methods, generally known as water vapor tracers (WVTs) are based on coupling a moisture tagging technique with a global or regional meteorological model (*Singh et al.*, 2016; Insua-Costa and Miguez-Macho, 2017; *Eiras-Barca et al.*, 2017). This strategy allows the model to explicitly account for all physical processes affecting atmospheric moisture, but on the other hand, it cannot be run offline and thus cannot be coupled to an atmospheric reanalysis, for example. This paper presents a comparison between both Lagrangian methods for the landfalling ARs episode on November 2016 and

May 1994 described above. Two Lagrangian models will be considered depending on whether inertial forces on tracers are considered or not.

## 2   Inertial and Lagrangian Models

The atmospheric transport has been studied using wind field data retrieved from the European Center for Medium-Range Weather Forecast reanalysis, ERA-Interim (*Dee et al.*, 2011). The spatial resolution of the data set is approximately 80 km

(T255 spectral) on 60 vertical levels from the surface up to 0.1 hPa, and a temporal resolution of 6 hours. Datasets were retrieved in a longitude-latitude-pressure coordinate system $(\phi, \theta, P)$ on model levels, that were translated into a terrain-following coordinate system.



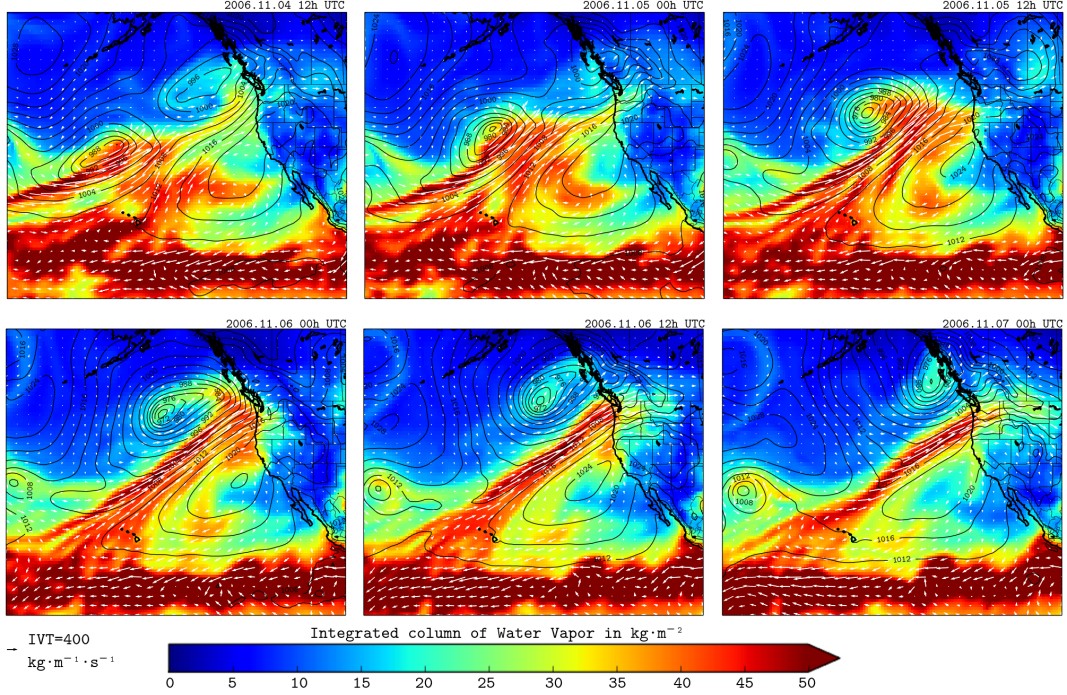

**Figure 1.** Composite analysis of the vertically integrated horizontal water vapor transport (IVT, kg m$^{-1}$ s$^{-1}$) and the integrated column of water vapor (IWV, kg m$^{-2}$) derived from the ERA-Interim daily reanalysis dataset for 4-7 November 2006.

Two types of fluid particles have been considered in this study; inertial and Lagrangian particles. In the first case, the particles' volume is assumed to change with time, while the Lagrangian particles keep their volume constant. A Lagrangian particle is then advected using the trajectory equation

$$\frac{dx_i}{dt} = u_i\left[x_i(t), t\right], \tag{1}$$

5 where $i$ is the $i$-component of the fluid velocity, and $u_i$ is the wind velocity interpolated in space and time from an external source at the particles' position $x_i$. Thus, Lagrangian particles follow wind stream trajectories. However, inertial tracers accelerate due to external forces acting on the particle and their motion in non-uniform incompressible flows can be modeled by the momentum equation (*Zapryanov and Tabakova*, 1999; *Michaelides*, 2003; *Takemura and Magnaudet*, 2004; *Pérez-Muñuzuri*, 2015; *Pérez-Muñuzuri and Garaboa-Paz*, 2016),

$$
\begin{aligned}
10 \quad \frac{dv_i}{dt} &= \frac{Du_i}{D_t} + 2\Omega \times U_i + C_L\left(U_i \times \omega_i\right) + \frac{9\nu}{R^2}U_i + \\
&\quad \frac{1}{2R^3}\left(\frac{d\left(R^3 U_i\right)}{dt} + 2R^3\frac{dU_i}{dt}\right) - \frac{9}{2R^3}\sqrt{\frac{\nu}{\pi}}\int_{-\infty}^{t}\frac{1}{\sqrt{\int_{\tau}^{t'}R^{-2}dt'}}\frac{d\left(RU_i\right)}{d\tau}d\tau,
\end{aligned}
\tag{2}
$$

where $v_i$ is the velocity of the inertial tracer, $u_i$ that of the fluid, $U_i = u_i - v_i$ the relative velocity between the fluid and the tracer, $\omega_i$ the fluid vorticity, $C_L = 0.5$ the lift coefficient for a sphere, $\rho$ the air density, $\nu = \mu/\rho$, the kinematic viscosity, and





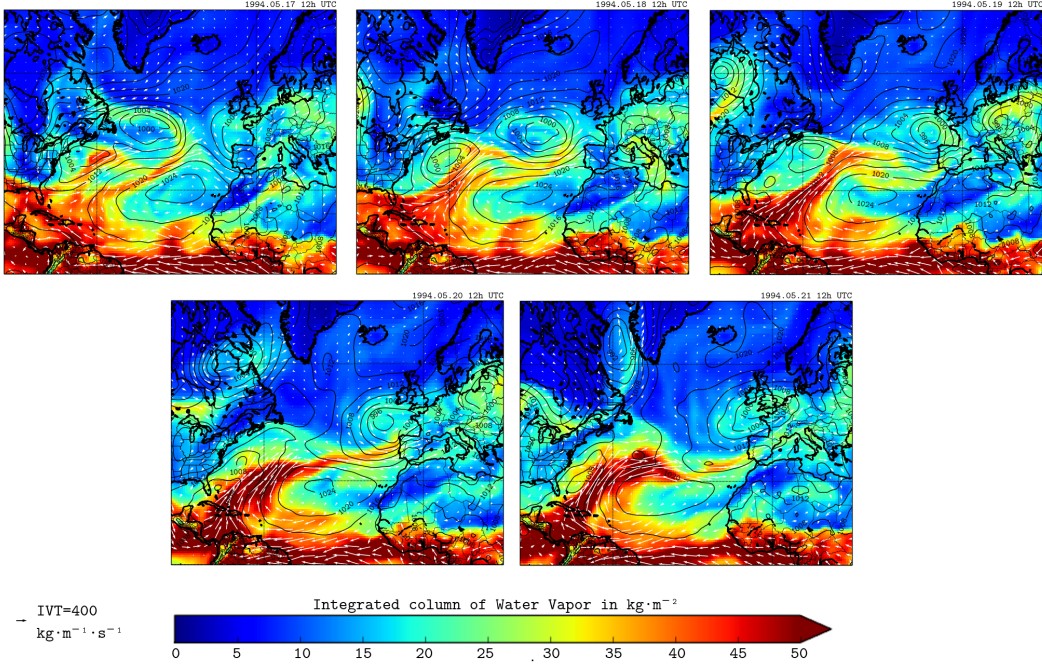

**Figure 2.** Composite analysis of the vertically integrated horizontal water vapor transport (IVT, kg m$^{-1}$ s$^{-1}$) and the integrated column of water vapor (IWV, kg m$^{-2}$) derived from the ERA-Interim daily reanalysis dataset for 17-21 May 1994.

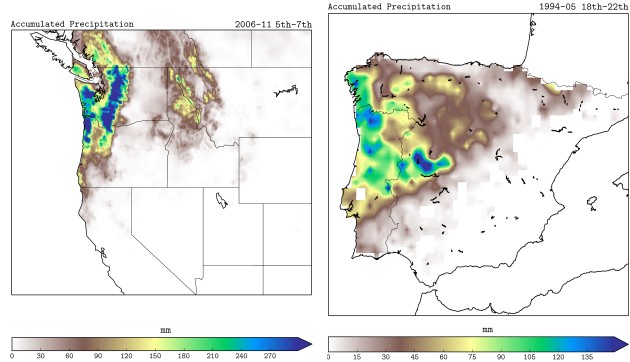

**Figure 3.** Accumulated precipitation for (a) 5-7 Nov 2006 and (b) 18-22 May 1994 in the western coast of United States and the Iberian Peninsula, respectively. Precipitation datasets were obtained from *Livneh et al.* (2015), *Herrera et al.* (2012) and *Belo-Pereira et al.* (2011).

$\Omega$ the Earth angular velocity. The six terms on the right represent, respectively, the force exerted by the undisturbed flow, the Coriolis force, the lift force, the Stokes drag, the viscous force, and the history force. In the last two terms, the effect of a spherical tracer with a time-dependent radius $R(t)$ has been considered (*Magnaudet and Legendre*, 1998; *Takemura and*





*Magnaudet*, 2004). The derivative $D/Dt$ is taken along the path of a fluid element, whereas the derivative $d/dt$ is taken along the trajectory of the particle.

The instantaneous parcel radius $R(t)$ is calculated from the Rayleigh–Plesset equation (*Plesset and Prosperetti*, 1977) of parcel dynamics

$$5 \quad R\frac{d^2R}{dt^2} + \frac{3}{2}\left(\frac{dR}{dt}\right)^2 = \frac{P_f - P}{\rho_f}. \qquad (3)$$

Further generalizations of this equation to a compressible fluid (*Prosperetti*, 1987) have been published, but for the purpose of this study we will keep on a first order approach, since the bubbly flow is mainly driven by the momentum equation (2). In Eq. (3), $dR/dt$ and $d^2R/dt^2$ are the parcel wall velocity and acceleration, respectively, $P_f$ is the pressure in the fluid at the parcel interface, and $P$ is the pressure field imposed by the flow. The pressure at the parcel interface is given by,

$$10 \quad P_f(R) = P_v + P_g - \frac{2\gamma}{R} - \frac{4\mu}{R}\frac{dR}{dt}, \qquad (4)$$

in which the first two terms are the internal pressure of the parcel related to the partial pressure due to vapor content $P_v$ and gas content $P_g$, respectively, and the last terms account for the interface curvature effect and the viscous stress at the interface. Surface tension is given by $\gamma$ and for our simulations can be considered negligible $\gamma \approx 0$. The gas pressure inside the parcel changes as the parcel contracts or expands. As the total amount of gas in the tracer remains constant, the tracer radius and gas pressure are related by $P_g = P_{g0}\left(R/R_0\right)^{3\alpha}$, where $\alpha = 1$ for an isothermal process, or equal to the ratio of specific heats for an adiabatic process. The external $P$ and vapor pressures are interpolated in space and time from the meteorological model at the particle position.

Updrafts and downdrafts due to moist convection were considered for both models. To represent convective transport in a particle dispersion model, it is necessary to redistribute particles in the entire vertical column as these transports are not represented by the ERA-Interim vertical velocity. Here, we follow the same convective parameterization implemented in FLEXible PARTicle dispersion model (FLEXPART) (*Emanuel and Zivkovic-Rothman*, 1999; *Stohl et al.*, 2005). Mesoscale wind fluctuations not solved by the ECMWF data are included in Eqs. (1-2) as a Gaussian random term with variance equals to the variance of the wind at the grid scale (*Stohl et al.*, 2005).

For the numerical experiments, a regular grid of $N = 80 \times 50$ particles is homogeneously distributed in the intervals $(\theta, \phi) \in [160W, 110W] \times [7N, 30N]$ (Pacific AR) and $(\theta, \phi) \in [60W, 20W] \times [15N, 30N]$ (Atlantic AR), and for 40 vertical levels from the surface up to 15 km above the ground. Then, 3D Lagrangian simulations have been performed so that particle trajectories are computed integrating equations above using a 4-th order Runge-Kutta scheme with a fixed time step of $\Delta t = 300$ s, and multilinear interpolation in time and space from current 60-level ECMWF data. Particles are advected during 120 hours beginning the 3th November 2006 (Pacific AR) and 16th May 1994 (Atlantic AR) at 0h UTC and every 6 hours a new grid of particles is released from the original location. The history term in Eq. (2) is integrated following the numerical integration scheme depicted by *Daitche* (2013).

To study the trajectory followed by a tagged mass of vapor, an initial volume of radius $R_0 = 5$ m was used. Different $R_0$ values, ranging from 1 m to 500 m, were also considered without affecting significantly the results shown below. Decreasing





the tracers radius, the inertial effects diminish and the results approach that of the Lagrangian particles. Initially, a specific humidity $q_v^T$ is assigned to each inertial/Lagrangian particle. The net change of water vapor content is given by,

$$e - p = \frac{d(mq_v^T)}{dt} \tag{5}$$

where $m$ is the mass of a particle, and $e - p$ measures the net excess or shortage of water vapor at the particle position. For the

inertial tracers, the volume of the particle changes with time, while $m$ is constant for the Lagrangian particles. At any time, we assume that water vapor and temperature of the particles are equal to the surrounding values interpolated from ERA-Interim at the tracer position $q_v^{part}(t) = q_v(t)$. Besides, the water vapor content inside the particle is equal to the tagged humidity plus some moisture up to $q_v$, $q_v^{part}(t) = q_v^T(t) + q_v^r(t)$. For $t = 0$, $q_v^{part}(t=0) = q_v^T(t=0)$ and $q_v^r(t=0) = 0$. Integration of Eq. (5) results in a decrease of the tagged moisture only when the water vapor excess $\varepsilon = q_v(t - \Delta t) - q_v(t)$ is positive,

$$q_v^T(t) = q_v^T(t - \Delta t) - \frac{\varepsilon \Delta t}{m} \left( \frac{q_v^T(t - \Delta t)}{q_v(t - \Delta t)} \right) \tag{6}$$

where $m = \rho V(t)$ for an inertial particle, and the last term represents the percentage of moisture reduction for the tagged water vapor. Otherwise, if $\varepsilon \leq 0$ the tagged moisture does not change.

## 3   Results

Tagged moisture advected from the Tropics and simulated with both Lagrangian models are shown in Figs. 4-9. For the Pacific

case, the integrated water vapor shows an intense plume of moisture extending from the tropical water vapor reservoir to Washington and Oregon as it was shown in the reanalysis, Fig. 1. Landfalling of the AR occurs the 6th of November at 0h UTC. During the next hours moisture continues to reach the continent displacing to the south and reaching North California. Inertial and Lagrangian tagged tracers are initially trapped by the vortical structure of the depression located northward of the AR that drags them while moving northeastern (see Fig. 1 for a time sequence). Compared to Lagrangian particles, a

crowded cloud of inertial particles loaded with moisture is observed around the depression. The AR is well defined for the inertial model while the pure Lagrangian one seems to lose quickly the tagged moisture from the Tropics. Note that 120h after being initialized, tagged inertial particles reach northern California favored by the northward turn of the low as observed in the IVT analysis, but not the Lagrangian ones. Tagged inertial particles can also be observed inland north of Montana 84h and 96h after initialization, and in the north of the British Columbia coast while this is not the case for the Lagrangian tracers

(IVT images show also some moisture on the same locations). The extreme nature of the November 2006 precipitation event (Fig 3) is reflected in the extreme values of the tagged moisture content over the North Pacific coast (see for example panel corresponding to 84h in Fig. 4).

The vertical distribution of water vapor from the Tropics, along the inertial AR is shown in Fig. 6. In the root of the AR, most of the tropical moisture remains close to surface, while for the leading edge, the humidity tends to ascend in the vertical

column. Once the AR reaches the Pacific coast the tagged water vapor ascends due to the topography, and the moisture content diminishes inland as precipitation develops. These results are in agreement with those obtained by *Eiras-Barca et al.* (2017).





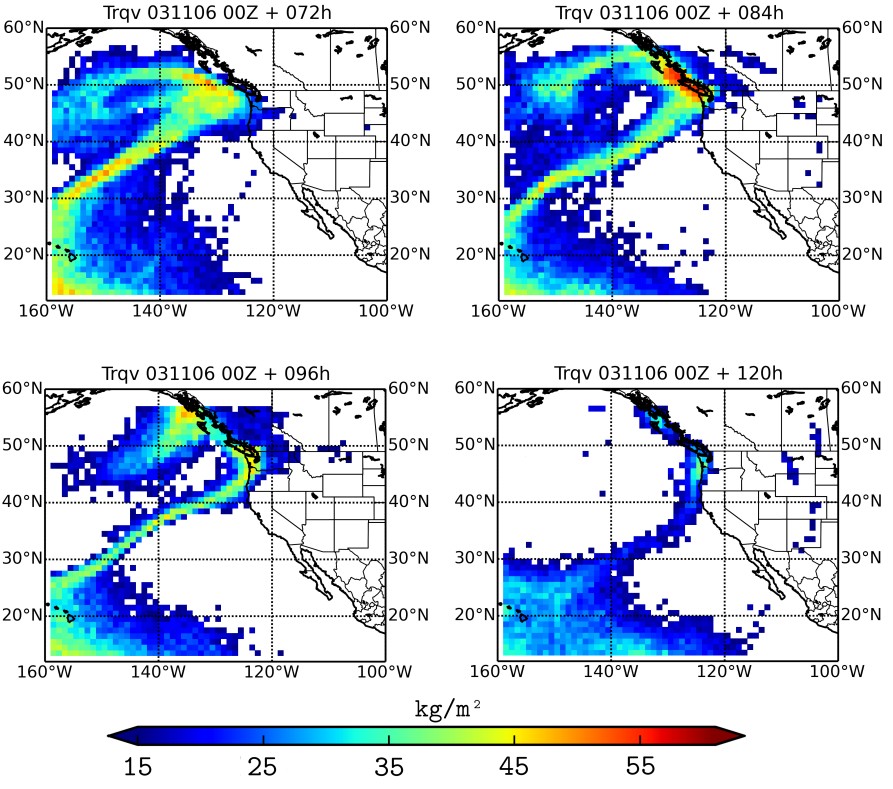

**Figure 4.** Vertically integrated tagged water vapor from the Tropics obtained from the inertial model for the November 2006 case.

Figure 7 resumes the temporal behavior of both inertial and Lagrangian moisture contents over three sites located in the coast of Washington, Oregon and North California states, respectively, compared to the IWV from the analysis. As mentioned above, the tagged vapor content of the inertial tracers fits well to the moisture content time series obtained from the analysis as the AR approaches Washington, and the trend is reproduced in the southern locations Oregon and North California. However,

5  the simulated concentration values are smaller in these two regions. On the other hand, Lagrangian particles moisture content increases rapidly with time reaching a maximum at approximately 40h after initialization and decreases to zero for the end of the simulation.

The next case of study corresponds to an Atmospheric River observed in 1994 landfalling in the western Iberian Peninsula on May 20th. Figures 8-9 reproduce the AR trajectory over the Atlantic ocean. The IWV analysis shown previously demonstrate the presence of two moisture branches reaching the Iberian Peninsula one after the other. This is observed in both Lagrangian

10  simulations; the first branch of the AR reaches Iberia approximately 72 hours after initialization (May 19th), while the second one landfalls the Peninsula on May 20th at 00Z. As it was observed in the previous case, the shape of the Lagrangian AR is more disintegrated than for the inertial simulations. In both cases, for the first hours of simulation, the presence of a depression





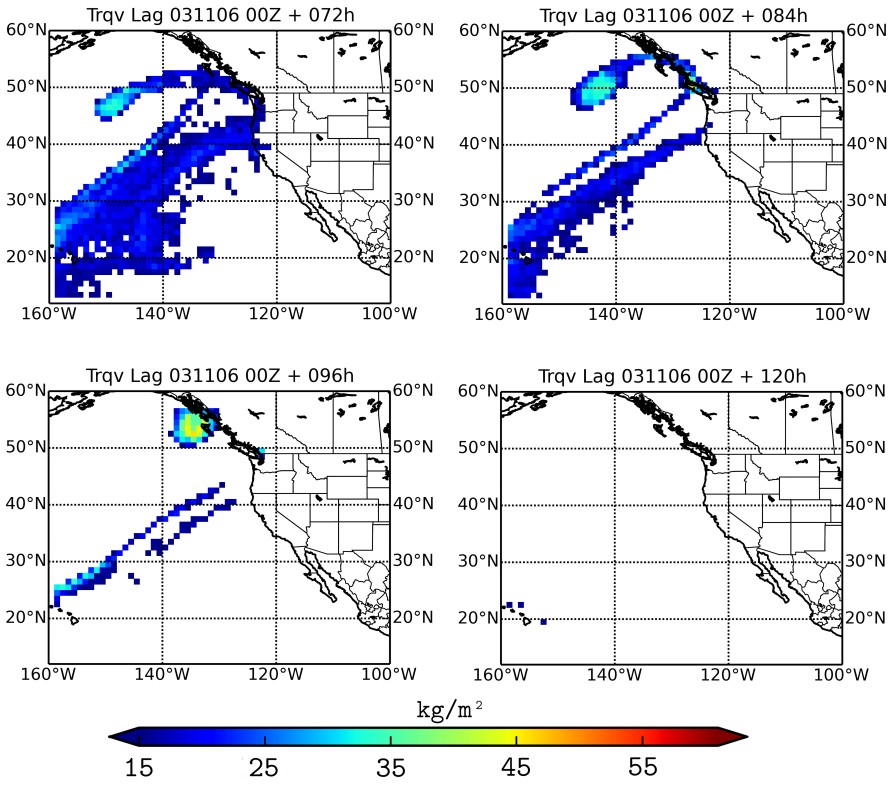

**Figure 5.** Vertically integrated tagged water vapor from the Tropics obtained from the Lagrangian model for the November 2006 case.

located northward of the plume (30W,40N) drags the particles towards the Iberian Peninsula. At the same time, a smaller depression (60W,43N) curls the tagged particles (second branch of the AR) reaching the Labrador Peninsula. The anticyclone in the middle of the Atlantic Ocean, clearly visible in the IWV images for May 20th at 12Z, is also reproduced as the inertial tagged particles curl around its center. Although the shape of the AR is clearly visible for both Lagrangian simulations, the

5  amount of moisture content that reaches Iberia is smaller than for the Pacific case previously analyzed. This translated into lesser precipitation rates.

Time evolution of the tagged vertically integrated moisture concentration is shown in Fig. 10 for three coastal sites located in the west of the Iberian Peninsula. Both AR branches described earlier are clearly visible from the IVW analysis data (triangles and red line) as local maxima of the moisture time series. Both peaks are delayed as the AR slides to the south from Galicia

10  (NW Spain) to Lisbon. Inertial particle simulation reproduces well the second peak and its evolution, although the moisture content for the Lisbon site is smaller than observed. However, the Lagrangian tracers reach the three sites for the first and second peak of the AR landfalling but not in intensity and the moisture content values obtained for the Lisbon site are smaller than for the inertial case.





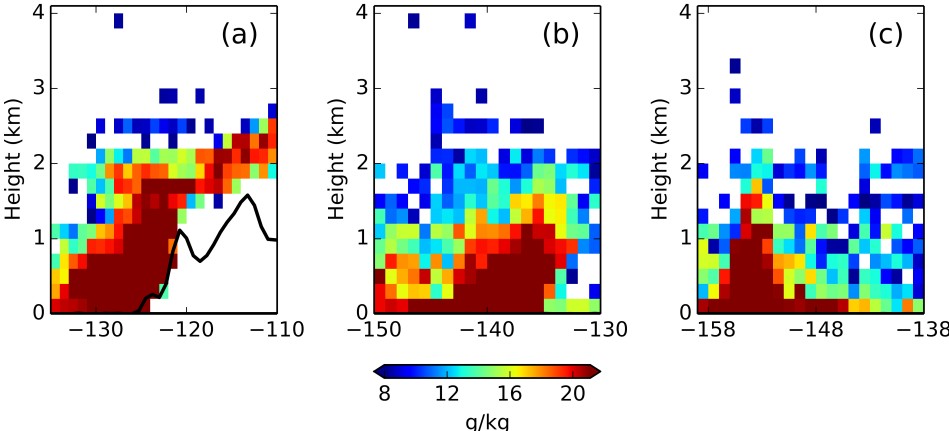

**Figure 6.** Transversal cross sections along the central axis of the atmospheric river in the Pacific basin at latitudes 48 (a), 38 (b), and 30 (c) for November 6, 12Z. The plots show the tagged inertial water vapor tracers $q_v$ in g/kg. Solid line in the left panel corresponds to the topography profile.

In both case studies, we note that for those tagged tracers reaching the continent, their vapor content diminish due to orographic ascent leading to precipitation extremes. Only for the Pacific region some tracers were observed inland (Montana State and west Canada) due to the high moisture and strong wind values observed in the simulations. On the other hand, we observed that inertial tracers keep their water vapor content longer than the Lagrangian ones near the low pressure areas.

The effect of tracer contraction and expansion is analyzed in Fig. 11 for both cases. To that end, the vertically integrated ratio $R(t)/R_0$ is represented for the same AR positions shown in Figs. 4-8. Note that the largest tracer volumes correspond to the highest values of the moisture content, due to uplift motions of the tracers inside the ARs and the Tropic region. On the other hand, the tracers volume rapidly decreases to zero inland due to the lost of waver vapor as precipitation takes place. Maximum volume values are attained near the Pacific coast as it was observed in the $q_v$ profiles described above.

## 4 Conclusions

Two Lagrangian and inertial models have been used to compare the trajectories of tagged moisture from the Tropics to evaluate the intense atmospheric river that devastated the Pacific North Western America with flooding rains and intense winds in early November 2006, and the AR that affected the western of the Iberian Peninsula during mid May 1994. Both models reproduce the structure of the ARs, but inertial tracers keep the moisture content longer and farther than the pure Lagrangian ones. Our results show that between 80% and 90% of the moisture observed near the coast has been advected by ARs, and has a tropical or subtropical origin. Local convergence transport is responsible for the remaining moisture (*Eiras-Barca et al.*, 2017). Comparing the inertial and Lagrangian models, the tagged inertial moisture concentrations over the Pacific coast are larger





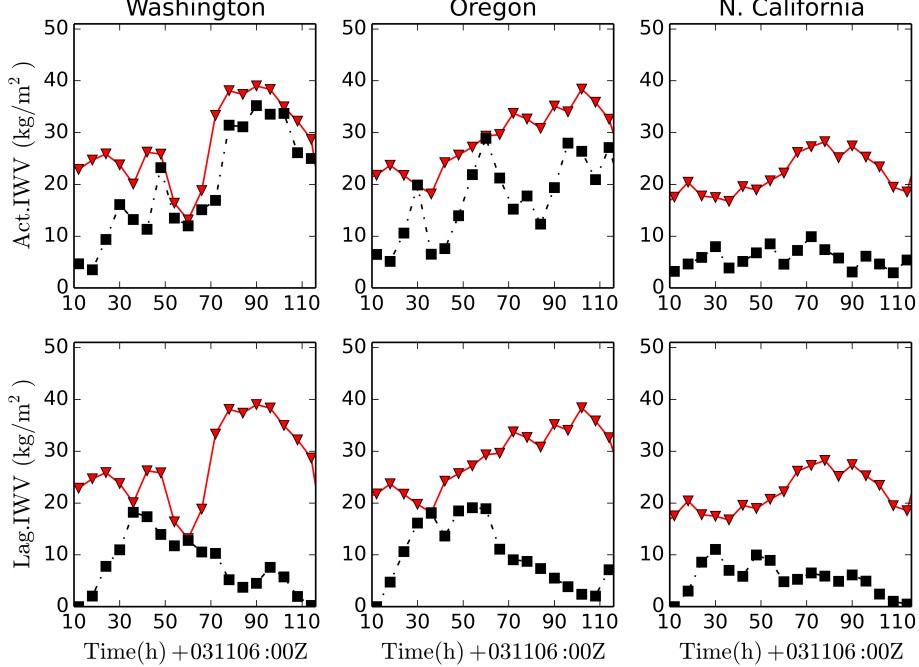

**Figure 7.** Time evolution of the tagged vertically integrated moisture concentration for three sites in the northeastern coast of US compared to the IWV obtained from the analysis (red triangles) for the November 2006 case. Inertial (upper row of panels) and Lagrangian (lower row of panels) tracers.

than for the Lagrangian ones and closer to the IWV analysis. For the Atlantic coast sites, the second branch of the river was correctly simulated by the inertial model, but not for the Lagrangian model.

The parameterization of Eq. (5) in terms of the vapor excess for the tagged tracers (6) has proven to be consistent with observations. The ability of the inertial tracers to respond to both thermodynamics and dynamical atmospheric changes has

5    proven to be an important issue to adequately describe the amount of water vapor content traveling long distances. The influence of a time-dependent radius is more important in the vicinity of the ARs as tracers move in the vertical direction. Although both Lagrangian and inertial models used the same wind fields and convective parameterization, the tagged moisture content at the coast was always smaller for the former model.

These results highlight the important contribution of tropics moisture to atmospheric rivers and the different dynamical

10   behaviors reproduced by the two models considered herewith. However, an in-depth investigation with a sufficient number of cases and further diagnostics would be needed to draw a more robust general conclusion.

*Competing interests.*  The authors declare that they have no conflict of interest.



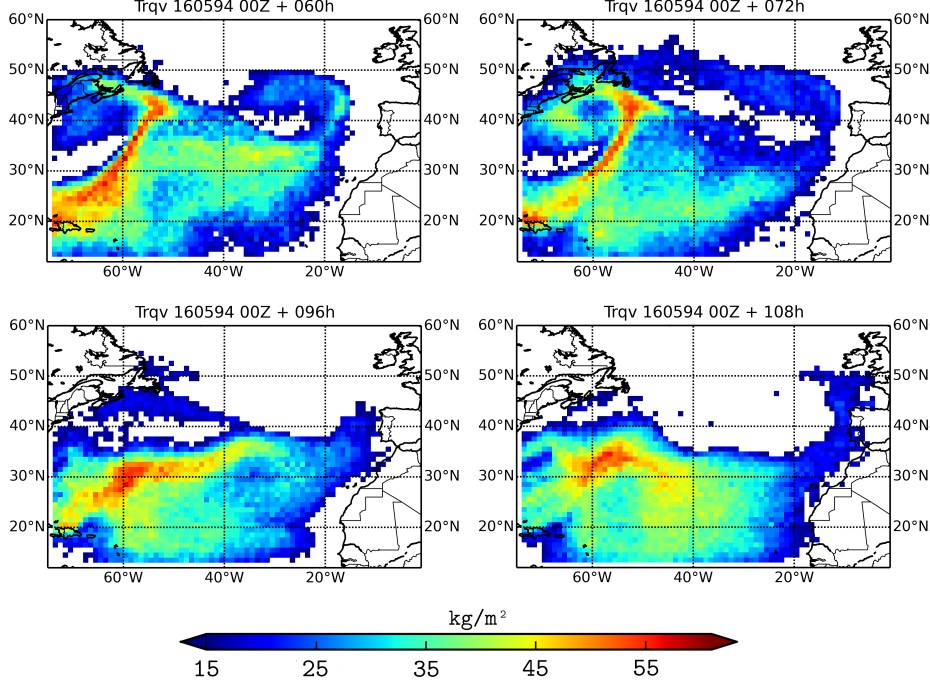

**Figure 8.** Vertically integrated tagged water vapor from the Tropics obtained from the inertial model for the May 1994 case.

*Acknowledgements.* ERA-Interim data were supported by ECMWF. This work was financially supported by Ministerio de Economía, Industria y Competitividad (CGL2017-89859-R and CGL2013-45932-R), and contributions by the COST Action MP1305 and CRETUS Strategic Partnership (AGRUP2015/02). All these programmes are co-funded by the ERDF (EU). Computational part of this work was done in the Supercomputing Center of Galicia, CESGA.





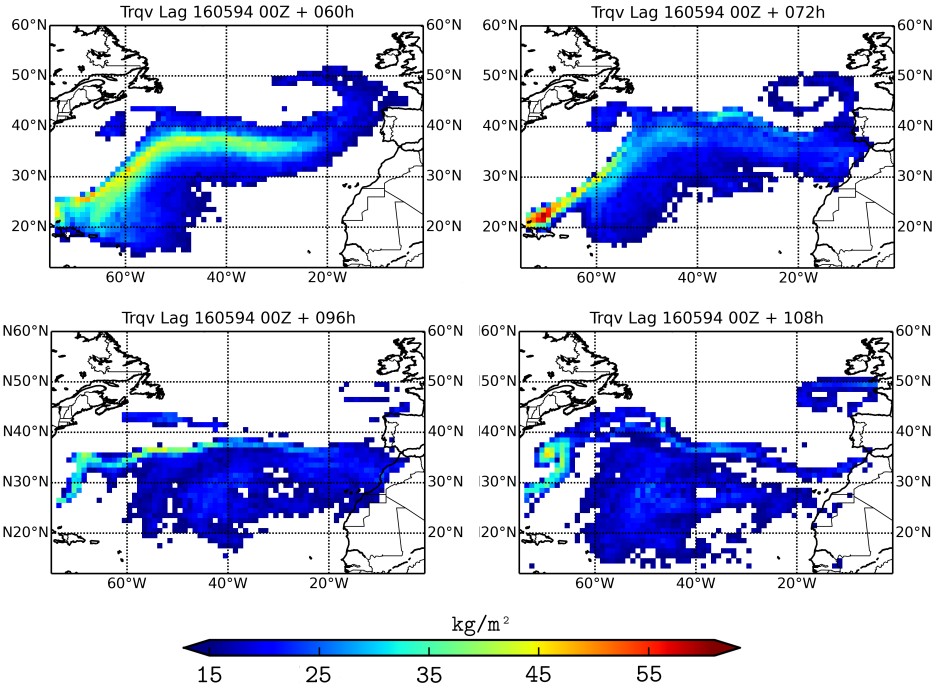

**Figure 9.** Vertically integrated tagged water vapor from the Tropics obtained from the Lagrangian model for the May 1994 case.

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





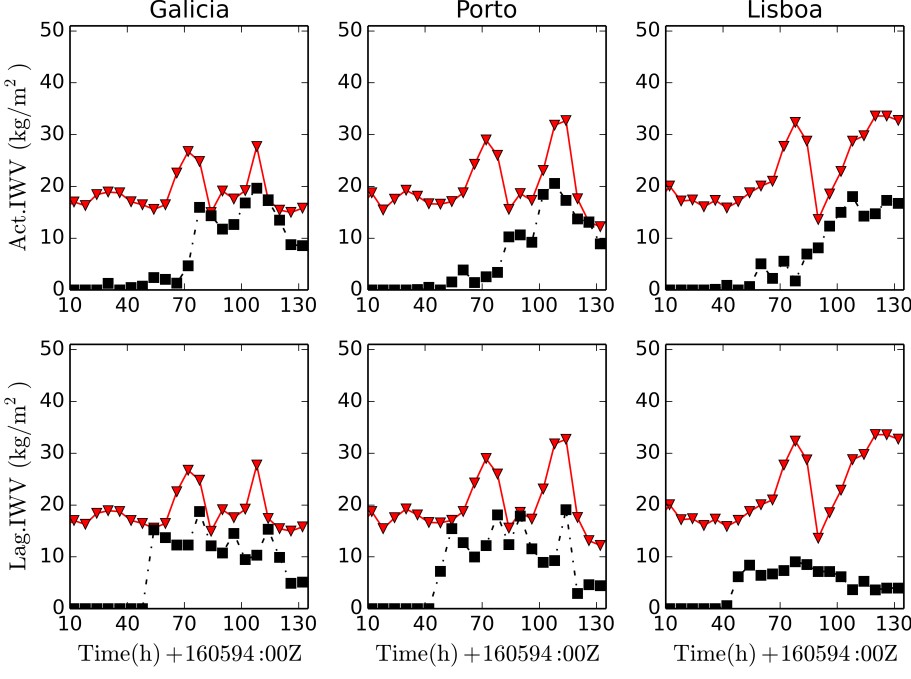

**Figure 10.** Time evolution of the tagged vertically integrated moisture concentration for three coastal sites in the western of the Iberian Peninsula compared to the IWV obtained from the analysis (red triangles) for the May 1994 Atlantic case. Inertial (upper row of panels) and Lagrangian (lower row of panels) tracers.

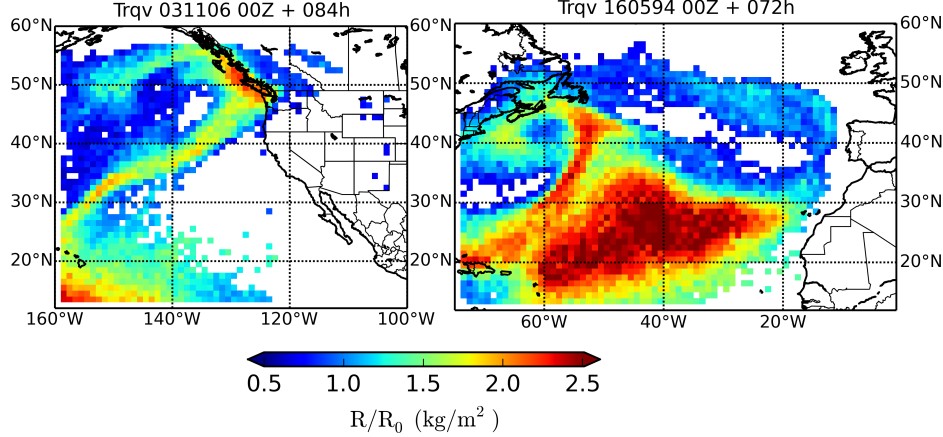

**Figure 11.** Vertically integrated $R/R_0$ for the Pacific (left) and Atlantic (right) oceans.





Eiras-Barca, J., Dominguez, F., Hu, H., Garaboa-Paz, A. D., and Miguez-Macho, G.: Evaluation of the Moisture Sources in two Extreme Landfalling Atmospheric River Events using an Eulerian WRF-Tracers tool, Earth Syst. Dynam. Discussions, 2017, 1-21, doi:10.5194/esd-2017-63, 2017.

Eiras-Barca, J., Ramos, A. M., Pinto, J. G., Trigo, R. M., Liberato, M. L., and Miguez-Macho, G. The concurrence of atmospheric rivers and explosive cyclogenesis in the North Atlantic and North Pacific basins. Earth System Dynamics, 9(1), 91, doi:10.5194/esd-9-91-2018, 2018.

Emanuel, K. A., and Zivkovic-Rothman, M. Development and evaluation of a convection scheme for use in climate models. J. Atmos. Sci., 56, 1766–1782, doi:10.1175/1520-0469(1999)056<1766:DAEOAC>2.0.CO;2, 1999.

Forster, C., Stohl, A., and Seibert, P., Parameterization of convective transport in a Lagrangian particle dispersion model and its evaluation. J. Appl. Meteor. Climatol., 46, 403–422, doi:10.1175/JAM2470.1, 2007.

Garaboa-Paz, D., J. Eiras-Barca, F. Huhn, and V. Pérez-Muñuzuri. Lagrangian coherent structures along atmospheric rivers, Chaos, 25(6), 063105, doi:10.1063/1.4919768, 2015.

Garaboa-Paz, D., J. Eiras-Barca, and V. Pérez-Muñuzuri. Climatology of Lyapunov exponents: the link between atmospheric rivers and large-scale mixing variability. Earth Syst. Dynam., 8, 865–873, doi:10.5194/esd-8-865-2017, 2017.

Gimeno, L., Drumond, A., Nieto, R., Trigo, R. M., and Stohl, A. On the origin of continental precipitation, Geophysical Research Letters, 37, 1–7, doi:10.1029/2010GL043712, 2010.

Gimeno, L., Stohl, A., Trigo, R. M., Dominguez, F., Yoshimura, K., Yu, L., Drumond, A., Durn-Quesada, A. M., and Nieto, R. Oceanic and terrestrial sources of continental precipitation, Reviews of Geophysics, 50, 1–41, doi:10.1029/2012RG000389, 2012.

Gimeno, L., F. Dominguez, R. Nieto, R. Trigo, A. Drumond, C.J.C. Reason, A. Taschetto, A.M. Ramos, R. Kumar, and J. Marengo. Major Mechanisms of Atmospheric Moisture Transport and Their Role in Extreme Precipitation Events. Annual Review of Environment and Resources, 41(1), 117–141, doi:annurevenviron-110615-085558, 2016.

Herrera, S., Gutiérrez, J.M., Ancell, R., Pons, M.R., Frías, M.D., and Fernández, J. Development and analysis of a 50-year high-resolution daily gridded precipitation dataset over Spain (Spain02). International Journal of Climatology, 32(1), 74–85, doi:10.1002/joc.2256, 2012.

Insua-Costa, D. and Miguez-Macho, G. A new moisture tagging capability in the Weather Research and Forecasting Model: formulation, validation and application to the 2014 Great Lake-effect snowstorm, Earth Syst. Dynam. Discuss., doi:10.5194/esd-2017-80, under review, 2017.

Lavers, D.A., and G. Villarini. The nexus between atmospheric rivers and extreme precipitation across Europe. Geophys. Res. Lett., 40(12), 3259–3264, doi:10.1002/grl.50636, 2013.

Livneh, B., Bohn, T. J., Pierce, D. W., Munoz-Arriola, F., Nijssen, B., Vose, R., Cayan, D. R., and Brekke, L.: A spatially comprehensive, hydrometeorological data set for Mexico, the US, and Southern Canada 1950–2013, Scientific data, 2, 150042, doi:10.1038/sdata.2015.42, 2015.

Magnaudet, J. and Legendre, D. The viscous drag force on a spherical bubble with a time-dependent radius. Phys. Fluids, 10, 550–554, doi:10.1063/1.869582, 1998.

Maxey, M.R. and Riley, J.J. Equation of motion for a small rigid sphere in a nonuniform flow. Phys. Fluids, 26, 883–889, doi:10.1063/1.864230, 1983.

Michaelides, E.E. Hydrodynamic force and heat/mass transfer from particles, bubbles, and drops. J. Fluids Eng. 125, 209–238, doi:10.1115/1.1537258, 2003.




Neiman, P.J., Ralph, F.M., Wick, G.A., Kuo, Y., Wee, T., Ma, Z., Taylor, G.H., and Dettinger, M.D. Diagnosis of an Intense Atmospheric River Impacting the Pacific Northwest: Storm Summary and Offshore Vertical Structure Observed with COSMIC Satellite Retrievals. Mon. Wea. Rev., 136, 4398–4420, doi:10.1175/2008MWR2550.1, 2008.

Newell, R.E., N.E. Newell, Y. Zhu, and C. Scott. Tropospheric rivers? A pilot study. Geophys. Res. Lett., 19(24), 2401–2404, doi:10.1029/92GL02916, 1992.

Pérez-Muñuzuri, V. Clustering of inertial particles in compressible flows, Phys. Rev. E, 91, 052906, doi:10.1103/PhysRevE.91.052906, 2015.

Pérez-Muñuzuri, V., and Garaboa-Paz, D. Mixing of spherical bubbles with time-dependent radius in incompressible flows, Phys. Rev. E, 93, 023107, doi:10.1103/PhysRevE.93.023107, 2016.

Plesset, M.S., and Prosperetti, A. Bubble dynamics and cavitation. Ann. Rev. Fluid Mech., 9, 145–185, doi:10.1146/annurev.fl.09.010177.001045, 1977.

Prosperetti, A. The equation of bubble dynamics in a compressible liquid. Phys. Fluids, 30, 3626–3628, doi:10.1063/1.866445, 1987.

Ralph, F. M., P. J. Neiman, D. E. Kingsmill, P. O. G. Persson, A. B. White, E. T. Strem, E. D. Andrews, R. C. Antweiler, and G. A. Wick. Satellite and CALJET aircraft observations of atmospheric rivers over the eastern North Pacific Ocean during the winter of 1997/98. Mon. Wea. Rev., 132, 1721–1745, 2004.

Ralph, F.M., and M.D. Dettinger. Storms, floods, and the science of atmospheric rivers. Eos T. Am. Geophys. Un., 92(32), 265–266, 2011.

Ramos, A.M., Nieto, R., Tomé, R., Gimeno, L., Trigo, R.M., Liberato, M.L., and Lavers, D. A. Atmospheric rivers moisture sources from a Lagrangian perspective. Earth Syst. Dynam., 7, 371–384, doi:10.5194/esd-7-371-2016, 2016.

Singh, H. A., Bitz, C. M., Nusbaumer, J., and Noone, D. C. A mathematical framework for analysis of water tracers: Part 1: Development of theory and application to the preindustrial mean state, Journal of Advances in Modeling Earth Systems, 8, 991–1013, 2016.

Stohl, A. and James, P. A Lagrangian analysis of the atmospheric branch of the global water cycle. Part I: Method description, validation, and demonstration for the August 2002 flooding in Central Europe, Journal of Hydrometeorology, 5, 656–678, doi:10.1175/1525-7541(2004)005<0656:ALAOTA>2.0.CO;2, 2004.

Stohl, A. and James, P. Lagrangian analysis of the atmospheric branch of the global water cycle. Part II: Moisture transports between Earths ocean basins and river catchments, Journal of Hydrometeorology, 6, 961–984, doi:10.1175/JHM470.1, 2005.

Stohl, A., Forster, C., Frank, A., Seibert, P., and Wotawa G. Technical Note: The Lagrangian particle dispersion model FLEXPART version 6.2. Atmos. Chem. Phys. 5, 2461-2474, doi:10.5194/acp-5-2461-2005, 2005.

Takemura, F., and Magnaudet, J. The history force on a rapidly shrinking bubble rising at finite Reynolds number. Phys. Fluids 16, 3247–3255, doi:10.1063/1.1760691, 2004.

Waliser, D., and Guan, B. Extreme winds and precipitation during landfall of atmospheric rivers. Nature Geoscience 10, 179, doi:10.1038/ngeo2894, 2017.

Zapryanov Z., and Tabakova, S. *Dynamics of bubbles, drops and rigid particles* (Science+Business Media Dordrecht, 1999).

Zhu, Y., and Newell, R. A proposed algorithm for moisture fluxes from atmospheric rivers, Mon. Wea. Rev., 126, 725–735, doi:10.1175/1520-0493(1998)1263C0725:APAFMF3E2.0.CO;2, 1998.