# Peer review of "Tagging moisture sources with Lagrangian and inertial tracers: Application to intense atmospheric river events"

_Earth System Dynamics, 2018_

## Referee Comment (RC1) · A. M. Durán-Quesada (Referee) · 20 Feb 2018

**Tagging moisture sources with Lagrangian and inertial tracers: Applications to intense atmospheric river events**

**Review ESD/10.5194/esd-2018-8**

*February 20, 2018*
* * *
**OVERVIEW**

The manuscript presents an analysis of atmospheric rivers (AR) comparing the performance of two tracers tools to diagnose the occurrence of AR events and provide further detail on the evolution of the systems. The method followed is properly fundamented and well described, inspite the simplicity of the approach it is based on a solid base for transport diagnosis in fluids. The results remark the role of moisture exports for the AR, however this is a very well known result and is not fully stated in the introduction (is mentioned at the end but lacking detail). The evaluation of the results supports that active tracers show better skills compared to Lagrangian tracing techniques, which is comparable to previous results obstained by the authors. From this perspective it is suggested to make clear and sound in the manuscript how the analysis of AR benefits from the use of one or other technique. For example, is it worth to use for AR climatology analysis, or whether the results support the use of IVT for forecasting. The manuscript still needs more effort in the presentation of results and their interpretation, the recomendation is to make a detailed revision of the manuscript to include major corrections, re-write the sections and improve the analysis.

**SPECIFIC COMMENTS**

1. The introduction requires a more coherent structure, it provides an introduction to AR, briefly introduces the case analysis and jumps to the introduction of analysis methods to finally include the role of tropical moisture exports for AR. As it is now is disorganized and hard to read. I suggest to rewrite the introduction with a better defined structure e.g a) ARs (what they are, how they benefit from tropical moisture and why are important in terms of heavy rainfall), b) analysis methods and key previous results and c) what new approach is proposed in the study.

2. Section 2 is well written, the methods are described in very good detail to ensure reproducibility. I recommend to include a section to present the case analyses, a formal synoptic description of the events and if possible information of the effect (e.g rainfall accumulated and rainfall rate during the AR life cycle) so that this piece of information can be considered for the analysis of the results. Figure 1-3 are poorly described and that may affect the interpretation of the results or at least their relevance.

3. Paragraph 20: Explain why tagged moisture is lost quickly from the pure Lagrangian model and what implication this have on the representation of the AR evolution.

4. Orographic ascend is a good mechanism to explain the fast moisture decrease for the Pacific case, however this mechanism is not comparable for the Atlantic, since the land configuration is very different and the pressure effect casused by the US topography is absent for the Atlantic case. Which mechanism is proposed for the Atlantic case?

5. Conclusions section is rather poor, the main result reported is a finding known from a previous research and it is not clear what is new from the present manuscript. The discussion lacks explanations on processes or how the results might provide new tools for climatology ARs analysis or even forecast support. The entire section must be re-written and pinpoint the main findings with a better justification.

---

## Referee Comment (RC2) · Anonymous Referee #2 · 19 Mar 2018

The paper presents two Lagrangian methods used to study atmospheric river events. In general the paper is written in poor language and grammar, it lacks explanations and validations. The conclusions are not demonstrated clearly in the text.

Specifically, the paper lacks references in the introduction and comparison to similar studies. The paper needs to be rewritten to improve the language and correct grammar, the text should include references to the figures included in the text and the figure captions should be expanded to describe the figures in more detail.

In Section 2 the assumptions used for the variables in the equations should be explained in relation to the context of the paper. Figure 7 and 10 should include legends,

all figures should have titles, and the figure caption for Figure 11 and similar figures should be expanded. In addition, calculations of the percentages presented in the conclusion should be described and shown in the results section along with validation the results.

---

## Author Comment (AC1) · 6 May 2018

We would like to thank both reviewers for their valuable comments and critics that we tried to take into account in the revised version of the manuscript. Hopefully, all the major and minor corrections pointed out by the reviewers have been corrected now. A detailed answer follows below. We provide replies to the reviewers' comments in bold. As well, corrections included in the manuscript are marked in red.

**Answer to Referee 1**

**(...) For example, is it worth to use for AR climatology analysis, or whether the results support the use of IVT for forecasting? The manuscript still needs more effort in the presentation of results and their interpretation, the recommendation is to make a detailed revision of the manuscript to include major corrections, re-write the sections and improve the analysis.**

With regard to the first question raised by the reviewer: In this manuscript we show that active tracers can explain the transport of the moisture better than classic Lagrangian ones. Nevertheless, this is not intended to identify ARs themselves. Thus, conclusions obtained in this analysis had no meaning unless the objective is to analyze the origin of the moisture/advection, but nothing related to create a climatology of the phenomenon, or to replace the use of IVT in the large variety of algorithms of detection proposed in the literature.

In order to follow the recommendations given by the referee, we have improved the description of the events and the interpretation of the results. We have included a supplementary material which addressed some concepts which were unclear. Referees can find all the improvements highlighted in red color.

**SPECIFIC COMMENTS**

**The introduction requires a more coherent structure; it provides an introduction to AR, briefly introduces the case analysis and jumps to the introduction of analysis methods to finally include the role of tropical moisture exports for AR. As it is now is disorganized and hard to read. I suggest to rewrite the introduction with a better defined structure e.g. a) ARs (what they are, how they benefit from tropical moisture and why are important in terms of heavy rainfall), b) analysis methods and key previous results and c) what new approach is proposed in the study. Section 2 is well written, the methods are described in very good detail to ensure reproducibility. I recommend to include a section to present the case analyses, a formal synoptic description of the events and if possible information of the effect (e.g. rainfall accumulated and rainfall rate during the AR life cycle) so that this piece of information can be considered for the analysis of the results. Figure 1-3 are poorly described and that may affect the interpretation of the results or at least their relevance.**

The introduction gives a synopsis of the main findings of atmospheric rivers in the literature.

Next, some discussion has been added concerning the tropical moisture export for the AR and the main concerns related to the source of water vapor feeding the rivers. It follows an explanation of the two AR events used in this paper. Supplementary information has also been added including precipitation rates and the 500 hPa geopotential height for both events. This information helps to better understand the ARs behavior. Finally, the section ends with a description of the existing methods and published papers, Lagrangian and Eulerian, to track moisture in the atmosphere. A glimpse of the inertial tracer model compared to the Lagrangian model is also explained there.

**Paragraph 20: Explain why tagged moisture is lost quickly from the pure Lagrangian model and what implication this have on the representation of the AR evolution.**

This is a good question. Both models, Lagrangian and inertial, have the same moist convection and condensation parameterizations (see Section 2). Thus, the difference between both results is due to the forces acting on the particle. The Lagrangian tracers just follow the streamlines, while the active tracers are accelerated due to different forces acting on them. Thus, vertical motion is not the same for both models. As a consequence, we suggest that Lagrangian particles suffer an overestimation of the vertical displacement leading to a rapid moisture loss.

**Orographic ascend is a good mechanism to explain the fast moisture decrease for the Pacific case, however this mechanism is not comparable for the Atlantic, since the land configuration is very different and the pressure effect caused by the US topography is absent for the Atlantic case. Which mechanism is proposed for the Atlantic case?**

We agree with the referee that orographic ascent is enhanced in the Pacific case when compared to the Atlantic one (added to the text). However, orographic ascent is also a key mechanism in the Atlantic coast of the Iberian Peninsula (Eiras-Barca, 2017). This, added to the natural wind ascent of the lead part of the AR associated to the low, explains the condensation of the moisture.

**Conclusions section is rather poor, the main result reported is a finding known from a previous research and it is not clear what is new from the present manuscript. The discussion lacks explanations on processes or how the results might provide new tools for climatology ARs analysis or even forecast support. The entire section must be re-written and pinpoint the main findings with a better justification.**

We are not sure to understand properly the first statement of the referee. As far as we know there are no previous simulations using active tracers instead of Lagrangian ones to track atmospheric moisture. However, we agree with the reviewer that probably we did not emphasize well enough these findings, so we have re-written the section.

---

## Author Comment (AC2) · 6 May 2018

We would like to thank both reviewers for their valuable comments and critics that we tried to take into account in the revised version of the manuscript. Hopefully, all the major and minor corrections pointed out by the reviewers have been corrected now. A detailed answer follows below. We provide replies to the reviewers' comments in bold. As well, corrections included in the manuscript are marked in red.

**Answer to Referee 2**

**The paper presents two Lagrangian methods used to study atmospheric river events. In general the paper is written in poor language and grammar, it lacks explanations and validations. The conclusions are not demonstrated clearly in the text. Specifically, the paper lacks references in the introduction and comparison to similar studies. The paper needs to be rewritten to improve the language and correct grammar, the text should include references to the figures included in the text and the figure captions should be expanded to describe the figures in more detail.**

The Introduction has been extended to incorporate *(i)* references on the tropical moisture export, and *(ii)* new figures (Suppl. Info.) concerning the precipitation rate for both case studies. New references have been added to compare with previous model simulations of ARs. We have double checked the English and grammar, and all figures are cited in the text.

Model validation has been done in terms of the IWV obtained from the analysis (Figs. 7 and 10). Unfortunately, there are no other sources for validation. Moreover, the IWV obtained from the analysis accumulates, not only that vapor coming from the Tropics, but also any other vapor from other sources, so a precise validation is uncertain.

**In Section 2 the assumptions used for the variables in the equations should be explained in relation to the context of the paper.**

Physical variables used in our model are now explained in the text. All of them have been related to atmospheric variables.

**Figure 7 and 10 should include legends, all figures should have titles, and the figure caption for Figure 11 and similar figures should be expanded.**

All captions have been enlarged. Concerning the legends in figures 7 and 10, the used symbols are described in the caption, but we leave the Editor to consider what the best option for the journal is.

**In addition, calculations of the percentages presented in the conclusion should be described and shown in the results section along with validation the results.**

The sentence has been deleted. The reviewer was right. We could calculate the RMSE for the IWV simulated and observed but that value has no sense as observations correspond to the total amount of vapor, while in our case only the vapor transported from the tropics is represented. This is more obvious for the Atlantic case as during the first two days no vapor coming from

the tropics was obtained near the Iberian Peninsula (see Fig.10).